# Cytotoxic and Hemolytic Activities of Extracts of the Fish Parasite Dinoflagellate *Amyloodinium ocellatum*

**DOI:** 10.3390/toxins14070467

**Published:** 2022-07-08

**Authors:** Márcio Moreira, Lucía Soliño, Cátia L. Marques, Vincent Laizé, Pedro Pousão-Ferreira, Pedro Reis Costa, Florbela Soares

**Affiliations:** 1S2AQUA—Collaborative Laboratory, Association for a Sustainable and Smart Aquaculture, Av. Parque Natural da Ria Formosa s/n, 8700-194 Olhão, Portugal; marcio.moreira@s2aquacolab.pt (M.M.); catia.marques@ipma.pt (C.L.M.); vlaize@ualg.pt (V.L.); pedro.pousao@ipma.pt (P.P.-F.); prcosta@ipma.pt (P.R.C.); 2IPMA—Portuguese Institute for the Ocean and Atmosphere, EPPO—Aquaculture Research Station, Av. Parque Natural da Ria Formosa s/n, 8700-194 Olhão, Portugal; 3CCMAR—Centre of Marine Sciences, University of Algarve, Campus de Gambelas, 8005-139 Faro, Portugal; luciasolino@gmail.com; 4IPMA—Portuguese Institute for the Ocean and Atmosphere, Av. Alfredo Magalhães Ramalho, n° 6, 1495-165 Algés, Portugal

**Keywords:** fish pathology, ectoparasite, toxicological response, hemolysis, cell viability

## Abstract

The dinoflagellate *Amyloodinium ocellatum* is the etiological agent of a parasitic disease named amyloodiniosis. Mortalities of diseased fish are usually attributed to anoxia, osmoregulatory impairment, or opportunistic bacterial infections. Nevertheless, the phylogenetic proximity of *A. ocellatum* to a group of toxin-producing dinoflagellates from *Pfiesteria*, *Parvodinium* and *Paulsenella* genera suggests that it may produce toxin-like compounds, adding a new dimension to the possible cause of mortalities in *A. ocellatum* outbreaks. To address this question, extracts prepared from different life stages of the parasite were tested in vitro for cytotoxic effects using two cell lines derived from branchial arches (ABSa15) and the caudal fin (CFSa1) of the gilthead seabream (*Sparus aurata*), and for hemolytic effects using erythrocytes purified from the blood of gilthead seabream juveniles. Cytotoxicity and a strong hemolytic effect, similar to those observed for *Karlodinium* toxins, were observed for the less polar extracts of the parasitic stage (trophont). A similar trend was observed for the less polar extracts of the infective stage (dinospores), although cell viability was only affected in the ABSa15 line. These results suggest that *A. ocellatum* produces tissue-specific toxic compounds that may have a role in the attachment of the dinospores’ and trophonts’ feeding process.

## 1. Introduction

Dinoflagellates are commonly observed in aquatic ecosystems, where they play a central role as primary producers and consumers or form a symbiotic relationship with many invertebrates [1]. About 140 of the approximately 2000 known aquatic living species are parasites, mostly from invertebrates [2]. Five genera have been reported as fish parasites: *Amyloodinium*, *Piscinoodinium*, *Crepidoodinium*, *Ichthyodinium* and *Oodinioides* [3].

The dinoflagellate *Amyloodinium ocellatum* is the aetiological agent of a parasitic disease with a great impact on marine finfish production, amyloodiniosis. This parasitosis affects more than one hundred species of farmed fish and crustaceans in a broad geographic distribution [4,5] and is considered a serious impediment to several warm-water aquaculture species worldwide, with considerable economic losses for the producers [6,7]. *Amyloodinium ocellatum* is a dinoflagellate with a direct life cycle that comprises three stages (Figure 1). The tomont is the resting stage and consists of a cyst that develops after trophont detachment from the host [8,9,10]. Each tomont can produce up to 256 dinospores (infestive free-living stage) in 3 days at 25 °C, that are released to the water column, each one capable of infecting a new host. When they find a host, they attach to the gill or skin and become a trophont (parasitic stage), that grows until 120–150 µm, detaches from the gill and encysts again as a tomont and restarts the cycle [11,12]. This exponential production of parasites can cause fast and asymptomatic outbreaks that can lead to high morbidity and mortality in brackish and marine warm-water fish [13,14,15].

Fish mortalities caused by amyloodiniosis are usually attributed to anoxia (by the destruction of the gill tissue) during heavy infestations [16], osmoregulatory impairment [17] or opportunistic bacterial infections associated with the lesions caused by the parasite in the gills and skin of the host [18]. Little information is available about the mechanisms of infestation, fixation and feeding of *A. ocellatum*, or about physiological responses of the host; thus, disease outcomes are still poorly understood, not elucidating the fast and high morbidity and mortality caused by this parasite, and preventive actions are limited [19]. Amyloodiniosis has been associated with behavioural and physiological changes in several fish species. Diseased fish display lethargic and sluggish movements, and slow opercular movement when heavily parasitized [13,16]. Changes in several blood and physiological parameters were also observed in the first 24 h of *A. ocellatum* infestation, prior to major changes in the gill histopathology of gilthead seabream (*Sparus aurata*) or the white seabream (*Diplodus sargus*) [19,20,21]. These changes have also been observed in heavily parasitized fish from other farmed species, such as the yellowtail (*Seriola dorsalis*) [22]. This could indicate a physiological impairment of the host, possibly caused by some compounds produced by *A. ocellatum* [23]. In this regard, *A. ocellatum* is phylogenetically close to *Pfiesteria*, *Parvodinium* and *Paulsenella* genera [24], which are known to be toxin-producing dinoflagellates at the origin of mass mortalities in wild and cultured finfish and shellfish [25], and the possibility that *A. ocellatum* also produces toxin-like compounds was hypothesized. Their existence remains to be confirmed.

Chromatographic fractioning coupled to cell-based assays is a suitable approach to assess the action of toxins or toxin-like compounds [26,27,28], providing that in vitro cell systems and endpoints are adapted, e.g., cell systems should be derived from tissues known to be affected by the toxin or known to express receptors for the suspected toxin [29,30,31,32]. Furthermore, a rescue assay using agonists and antagonists for the toxin target(s) can provide critical clues on toxin mechanisms and structure [33,34,35]. This method has been successfully applied to the study of phycotoxins with the capacity to bind and block ionic channels such as maitotoxins (MTXs) and palytoxins (PLTXs) in erythrocytes [36,37,38]. A simple assay based on fish erythrocytes has also been used to characterize the haemolytic action of the toxins [39]. Erythrocytes express primordial ionic channels and electrogenic pumps such as Na^+^/K^+^ ATPase, as well as active co-transporters and antiporters, which are common targets of natural toxins [40,41]. Erythrocyte lysis has been used as a proxy for the presence of toxic compounds in extracts of a wide range of harmful dinoflagellates [42,43,44].

The objective of this work was to assess whether toxin-like compounds are present in extracts of the parasitic dinoflagellate *A. ocellatum*, using gilthead seabream (*Sparus aurata*) in vitro cell systems to assess cytotoxic effects, and fish erythrocytes to assess hemolytic effects. Extracts corresponding to the different stages of the parasite life cycle will be assessed.

## 2. Results

### 2.1. Amyloodinium ocellatum Life Stages Collection

The system assembled for the production of the parasite reached maturity in four weeks, allowing us a daily production of 4000 ± 200 tomonts per day (Figure 2), and a stable concentration of 2000 ± 500 dinospores mL^−1^ with the ability to attach to the gills of the fish host (Figure 3).

The daily examination of the tank water did not reveal any contamination by other parasites and/or ciliates during the whole duration of the experimental work.

### 2.2. Viability of Fish Cells Exposed to Fractions of Amyloodinium ocellatum Life Stages Crude Extract

Fractions 1 and 2 of dinospore crude extract significantly decreased the viability of ABSa15 cells by 42.44% and 26.60%, respectively (Figure 4), while other fractions had no effect. None of the dinospore fractions affected the viability of CFSa1 cells (Figure 4). Fraction 1 of the trophont crude extract significantly decreased the viability of ABSa15 by 74.52% and the viability of CFSa1 cells by 57.93% (Figure 4), but none of the other fractions affected cell viability in both fish cell lines. None of the fractions of the tomont crude extract affected the viability of ABSa15 or CFSa1 cells, but Fraction 2 was able to stimulate cell proliferation by 35.40% in CFSa1 cells (Figure 4). Note that Fraction 1 of the tomont extract was not assessed for its effect on cell viability due to the scarcity of the recovered material.

### 2.3. Hemolytic Effect of Amyloodinium ocellatum Crude Extract Fractions on Gilthead Seabream Erythrocytes

Fish erythrocytes incubated with tomont Fractions 1 and 2 showed the highest percentage of hemolysis while Fractions 3, 4 and 5 showed an average of 18.45, 10.02 and 3.87% of hemolysis effect, respectively. These results indicate that active compounds in the tomont extract elute faster in organic solvents, probably due to their hydrophobic nature. Ca^2+^ channel inhibitors displayed few counteracting effects, except for SKF96356, which partially inhibited Fractions 1 and 2 and almost completely suppressed the effect in Fractions 3, 4 and 5. Similar behavior was observed in Fraction 3 pre-incubated with KB-R7943, while this compound seemed to exhibit a synergistic effect in Fractions 4 and 5 (one-way ANOVA, Tukey’s post-test, *p* < 0.01). On the other hand, verapamil showed no inhibitory response in Fractions 1 to 5 (Figure 5). Although there were no significant differences observed between the three treatments and the control (two-way ANOVA, Bonferroni post-test, *p* < 0.01), these observations may point to the indirect involvement of membrane ionic calcium channels other than L-type VGCC. Nevertheless, the specific action of Fractions 1 and 2 on L-type and non-L-type VGCC or the antiporter NXC seems unlikely.

On the dinospore crude extract fractions, the highest hemolytic effect was observed in Fraction 2 (100%), followed by Fractions 1, 5, 4 and 3 (32.64, 13.54, 6.60, and 0.17% respectively; Figure 5). These results indicate a tendency for a faster elution of active compounds in the dinospore extract, probably due to their hydrophobic nature. Ca^2+^ channel inhibitors exhibited an antagonistic effect in Fractions 1 and 2, while a synergistic effect was observed in the most polar fractions (one-way ANOVA, Tukey’s post-test, *p* < 0.01) (Figure 5). When compared to the control, this effect was significant in the case of verapamil for all fractions and Fraction 1 in the case of SKF96365 and KB-R7943, respectively (two-way ANOVA, Bonferroni post-test, *p* < 0.01). The inhibitory/agonistic response of KB-R7943 was also significantly different to the control in Fractions 2 and 4 (*p* < 0.01).

Regarding the trophont crude extract fractions, the least hydrophilic Fractions 1 and 2 caused the highest percentage of hemolysis (99.71 and 88.95%, respectively, see Figure 5), which indicates that active compounds in the trophont extracts elute faster in organic solvents, probably due to their hydrophobic nature. This hemolytic effect could not be inhibited by any of the tested compounds for these two extracts (two-way ANOVA, Bonferroni post-test, *p* < 0.01), which clearly demonstrates a non-specific effect on erythrocytes. Although SKF96365 could not be tested in trophont fractions due to reagent scarcity, preliminary trials in our laboratory for the same species in the same conditions indicated a similar effect to those obtained with verapamil and KB-R7943, with 10.02% inhibition in Fraction 1 and 5.04% inhibition in Fraction 2.

## 3. Discussion

The presence of compounds with cytotoxic and hemolytic activities has been confirmed in crude extracts prepared from dinospores and trophonts of *A. ocellatum*, and compounds with a proliferative action may be present in tomont extracts. More specifically, fractions prepared from tomont crude extract did not affect the viability of fish cells, independently of their origin (i.e., branchial arches and caudal fin). Since the tomont is a resting stage, it is expected that the parasite is dormant and does not produce metabolites in a quantity that could trigger cytotoxic effects [10,17]. Interestingly, Fraction 2 triggered a proliferative effect in CFSa1 cells. Although this remains to be confirmed, growth factors involved in the division of tomont into tomites (the stage that gives origin to dinospores) and that are probably present in tomont extracts [45] may have the capacity to stimulate the growth of the fish cells. Surprisingly, Fractions 1 and 2 of the tomont extract triggered a high level of hemolysis. Tomonts are not supposed to be in contact with the gill or skin tissues in the fish [17], thus should not produce hemolytic compounds. Compounds produced during the preceding life stage, i.e., the trophonts, or compounds related to metabolic processes involved in the tomont division may be involved in the hemolytic effect. This would also explain the unlikeliness of the observed specific action of Fractions 1 and 2 on L-type and non-L-type VGCC or the antiporter NXC specific action.

Fractions 1 and 2 prepared from the dinospores extract specifically reduced the viability of the cell line established from seabream branchial arches. Structurally, the fish gill epidermis is constituted by one to four layers of cuboidal or squamous epithelial cells [46,47], and is substantially different from the fin epidermis, which is a stratified squamous epithelium similar to the fish skin [48,49]. Our in vitro data showing a stronger effect on the ABSa15 cells indicates that dinospores may have a higher specificity to gill epidermal cells. The partial inhibition of hemolysis observed when verapamil, SKF96365 and KB-R7943 were supplemented in Fractions 1 and 2 indicates the involvement of the three channels in the hemolytic action of the dinospores, although their role may be secondary and remains to be better understood. This resembles the hemolytic effect of MTXs on human and mouse red blood cells, suggesting the blockage of an unknown Ca^2+^ channel or mechanisms involving calmodulin activation pathways [36,50]. Previous studies have reported the reversal of membrane NXC exchangers as a secondary effect of Na^+^ influx in cells, leading to an increment of cytosolic Ca^2+^ [51]. The blockage of VGCC or NXC by verapamil, SKF96365 and KB-R7943 could result in a lower cytosolic Ca^2+^ and consequently lower hemolysis. On the other hand, the high synergistic effect observed in hydrophilic fractions in the presence of verapamil may point to the existence of a bioactive compound whose effect is counteracted in normal homeostatic conditions but is somehow dependent on L-type VGCCs. The effect of Ca^2+^ internal reservoirs may not be ruled out, since nucleated erythrocytes contain organelles that can contribute to cell osmosis regulation [40,41]. The decrease of Ca^2+^ influx when membrane Ca^2+^ channels are blocked may mobilize intracellular Ca^2+^ reservoirs and cause the ionic imbalance that leads to cell lysis. This strengthens the observation that dinospores may have a higher specificity for gill epidermal cells, possibly through the anchorage and fixation processes (see Figure 3) to these cells [23].

The cytotoxic effect of the less polar fraction of trophont extract on both cell lines may indicate the presence of compounds with the capacity to lyse cells, probably associated with digestive vacuoles present in the digestive area of the trophont [23,52], that could be related to the implementation of the trophont rhizoids on the cytoplasm of epithelial cells [45,53,54], as well as trophont feeding by the stomatopod structure [45,55] (see Figure 2B). The unspecific response of the trophont extract in the hemolytic assays suggests a mechanism of action similar to that of karlotoxins (KmTxs), which are synthesized by the dinoflagellates *Karlodinium* spp., a piscicide and closely related genus [56]. These findings strengthen the hypothesis placed by Cheung et al. [45] and Lom [23] that the stomatopod has been thought to inject histolytic substances into the host tissues and/or to be used for food ingestion.

## 4. Conclusions

In the present work, we demonstrated the existence of life stage-specific compounds secreted by the parasite *A. ocellatum* with the capacity to promote cell death. This can be related to parasite anchorage and fixation in the dinospore and with rhizoid establishment and feeding in the trophont stage (Figure 6). The non-specific response to the trophont extracts observed in the hemolytic test is similar to the karlotoxin, which could indicate a possible toxin-like compound production by *A. ocellatum*.

Further in vivo studies and other specific analyses to the *A. ocellatum* crude extract fractions of interest will be performed to elucidate if the *A. ocellatum* dinospores and trophonts produce ichthyotoxins, which could be of importance for the establishment of protocols regarding the commercialization of fish after an amyloodiniosis outbreak, due to the possible effects on humans of those putative ichthyotoxins.

## 5. Materials and Methods

### 5.1. Fish Culture Conditions

Gilthead seabream (*Sparus aurata*) juveniles, reared at the Aquaculture Research Station, National Institute for the Ocean and Atmosphere (EPPO-IPMA, Olhão, Portugal) following an in-house protocol [57]), were used for parasite production. A set of 200 juveniles weighing 80 to 100 g and with no history of ectoparasites, that were used to stimulate and maintain the infection and for parasite collection in the contamination tanks, were kept in 5 m^3^ fibreglass tanks in an open system and with a natural photoperiod. The water temperature was 23 ± 1 °C, and water salinity was 37.5–38 psu. Fish were fed with a commercial feed (Aquasoja Balance 5 mm, Sorgal^®^, S. João, Ovar, Portugal) twice a day, until satiation.

### 5.2. Establishment of Infestation Tanks for Parasite Production

Two 600 L rectangular fibreglass tanks filled with UV sterilized seawater were infested with 5000–6000 *A. ocellatum* tomonts collected from an induced outbreak according to the methodology described in [20,21]. Several naive and unparasitized gilthead seabreams juveniles were exposed to the parasite to increase the dinospore infective population of the tank and to maximize trophont and tomont production per day, as described in [19]. The water temperature was maintained at 24 ± 0.2 °C, 36.5 psu of salinity, and 100% dissolved oxygen in closed seawater systems, artificial aeration, and a 14 h light/10 h dark photoperiod. Five percent of the water in the system was replaced weekly with new sterile seawater to replace losses by evaporation in the infestation tanks. The concentration of dinospores in the water and the absence of contamination by other parasites and/or ciliates were accessed daily by microscopic observation according to the methodology described by Moreira et al. [19]. Two samples of water (490 mL) were collected in the middle of the water column and fixed with 10 mL of Lugol (5% of the total volume of the sample). The density of dinoflagellates was determined using the Utermöhl method for quantitative phytoplankton analysis [58]. The concentration of trophonts in the gills was accessed by microscopical observation of a wet mount of the two first right gill arches of the fish at 48 h post-infestation (hpi). The system was considered mature when the concentration of dinospores in the tanks reached 2000 dinospores mL^−1^ and the levels of trophont in the fish reached 5000–6000 trophonts per gill arch at 24 hpi. To avoid large fluctuations in the parasite load in the system, the necessary number of the different life stages of *A. ocellatum* for the experiment was collected two times per week, during a two-month period.

### 5.3. Amyloodinium ocellatum Life Stages Collection

#### 5.3.1. Tomont Collection

Tomonts were collected following an adaptation of the protocol established by Paperna [59], as described in [20]. Fish with jerky movements and darkened skin at 48 hpi were bathed in sterile distilled water for 1 to 2 min. Water was then filtered sequentially through a 500 μm mesh (to remove large-size debris), a 250 μm mesh (to remove intermediate-size debris), a 150 μm mesh (to remove small-size debris) and finally a 60 μm screen to concentrate and wash the tomonts. Tomonts were then concentrated, collected into a 15 mL falcon and then resuspended in 10 mL of UV sterilized seawater. A 1 mL sample was placed into a Sedgewick–Rafter chamber and counted in the microscope (Nikon^®^ Eclipse Ci, Nikon Instruments Europe, Amstelveen, The Netherlands) at 100× amplification. Tomonts were then concentrated again, cleaned with sterile distilled water to remove salts, resuspended in absolute ethanol and preserved at 4 °C until further use.

#### 5.3.2. Dinospore Collection

The water of the infestation tanks was sampled daily to determine the density of dinospores and to check for possible contaminations, as described in Section 5.2. When the dinospore load reached 8000 dinospores mL^−1^, water was filtered using 47 mm GE^®^ Whatman glass microfiber GF/C filters (pore size: 1.2 µm). The filters with dinospores were preserved in absolute ethanol at 4 °C until further use.

#### 5.3.3. Trophont Collection

After fish sacrifice with an overdose of 2-phenoxyethanol, according with the EU Directive 2010/63/EU for animal experimentation, the concentration of trophonts in the gills was accessed by microscopic observation of a wet mount of the two first right gill arches of the fish. After verification that the fishes presented between 50,000–100,000 *A. ocellatum* trophonts per gill arch, the first two left gill arches were collected from the infested fish, snap-frozen in liquid nitrogen and preserved at −80 °C until further use.

### 5.4. Production of Amyloodinium ocellatum Extracts by Solid-Phase Extraction Fractioning

Dry samples of dinospores (1 × 10^6^ dinospores per sample), tomonts (4 × 10^5^ tomonts per sample) and gills infested with trophonts (7.2 × 10^6^ trophonts per sample) were extracted with 3 mL of methanol (ACS reagent, ≥99.8%, Sigma^®^, St. Louis, MO, USA) by sonication for 3 min at 70% amplitude and 3 s pulses (Vibra-Cell VCX 500, Sonics & Materials Inc., Newtown, CT, USA) while cooling on an ice bath. Cell disruption was monitored under a microscope; when 90 ± 1% of the cells were disrupted the extracts were centrifuged at 1700× *g* for 10 min. The supernatants were collected and fractioned by Solid Phase Extraction (SPE) cartridges (Supelclean LC-18 SPE cartridge, 3 mL, Supelco, Sigma-Aldrich^®^, St. Louis, MO, USA), following the protocol previously described by Moeller et al., with modifications [27]. Briefly, cartridges were conditioned with 1 mL of 100% methanol and then, samples were loaded and fractionated using solvents of increasing polarity, where Fraction 1 corresponds to 100% methanol, Fraction 2 to 80% methanol, Fraction 3 to 50% methanol, Fraction 4 to 25% methanol and Fraction 5 to 0% methanol (100% water). The solvent was evaporated under vacuum at 60 °C before the hemolytic assays (RapidVap Vacuum Dry Evaporation system, Labconco, Kansas City, MO, USA).

### 5.5. Impact of Amyloodinium ocellatum Extracts on Fish Cell Viability

Gilthead seabream (*Sparus aurata*) cell lines ABSa15 (established from branchial arches) and CFSa1 (established from the caudal fin) were routinely cultured at 33 °C in Dulbecco’s modified Eagle’s medium (DMEM; Thermo Fisher Scientific^®^, Waltham, MA, USA) as described in Marques et al. [60] and Tiago et al. [61]. Cells were seeded in 96-well plates at a density of 1.5 × 10^3^ cells per well. After 24 h, the culture medium was renewed and supplemented with the extracts prepared as described in Section 5.4. at a final concentration of 1000 cell equivalent mL^−1^ for the tomont crude extract fractions, 500,000 cell equivalent mL^−1^ for the dinospore crude extract fractions, and 240,000 cell equivalent mL^−1^ for the trophont crude extract fractions. Cell viability was determined after 4 h of exposure using the XTT Cell Proliferation assay kit (Biotium, Fremont, CA, USA).

### 5.6. Hemolytic Assays

The hemolytic activity of the extracts was assessed using blood from healthy gilthead seabream reared in the tank used for parasite production. Red blood cells were collected by centrifugation at 240× *g* for 5 min at 15 °C. The supernatant was carefully removed, and red blood cells were resuspended in an equal quantity of PBS. This step was repeated until the PBS supernatant appeared clear. Erythrocytes were diluted in PBS to a final concentration of approximately 4.2–4.7 × 10^4^ cells mL^−1^. To evaluate the hemolytic activity of *A. ocellatum* extracts, 150 μL of each fraction was added to 150 μL of erythrocyte suspension. The mixture was gently homogenized and incubated for 24 h at 20 °C. A blank and positive control were performed with PBS and Milli-Q water (Millipore^®^, Burlington, MA, USA), respectively, corresponding to 0 and 100% hemolysis. Tubes were centrifuged at 480× *g* for 10 min and 150 μL of each supernatant was carefully transferred into a 96-well plate. Absorbance (A) was determined at 415 nm in a Sunrise microplate reader (Tecan^®^, Kawasaki, Japan). The percentage of hemolysis was calculated as a percentage (%), as described by Costabile [62].

To evaluate the specific effect of *A. ocellatum* fractions on calcium channels, the inhibitors Verapamil at 100 μM, SKF96365 at 100 μM and KB-R7943 at 29 μM were tested with each fraction. These compounds act on L-type voltage-gated calcium channels (L-VGCC), non-L-type VGCC and the Na^+^/Ca^2+^ exchanger (NXC), respectively. Since KB-R7943 is dissolved in DMSO, the hemolytic effect of this vehicle was also tested. Additionally, to assess the possible gill matrix effects in trophont fractions, an extract of non-infected gills was also tested. The concentrations that did not produce matrix effect were 24,000 trophont cells mL^−1^ for Fractions 1 and 2, and 240,000 trophont cells mL^−1^ for Fractions 3 to 5. All fractions were tested in the presence or absence of each inhibitor, and at concentrations similar to those that caused hemolysis for crude extracts. Due to the limited amount of some samples, the analysis of the dinospore fraction with SKF96365 wasn’t done, and only one concentration for each fraction and two replicates were performed.

### 5.7. Statistical Analysis

The normality of the data for the cell viability was assessed using the Shapiro–Wilk test and the homogeneity of variance was assessed using the Bartlett test. Since data were parametric, significant differences were determined through one-way ANOVA followed by Tukey’s post-test (*p* < 0.001). For the hemolytic tests, the normality was assessed using the Shapiro–Wilk test and the homogeneity of variance was assessed using the Bartlett test. Since the data were parametric, a two-way ANOVA followed by a Bonferroni post-test to detect differences between the tested parameters (*p* < 0.01). To access if the hemolytic tests had any synergistic or antagonistic effects, we performed a one-way ANOVA, followed by a Tukey’s post-test (*p* < 0.001). Statistical tests and graphics were performed using GraphPad Prism version 5.0.0 for Windows, GraphPad Software, San Diego, CA, USA, www.graphpad.com.

## Figures and Tables

**Figure 1 toxins-14-00467-f001:**
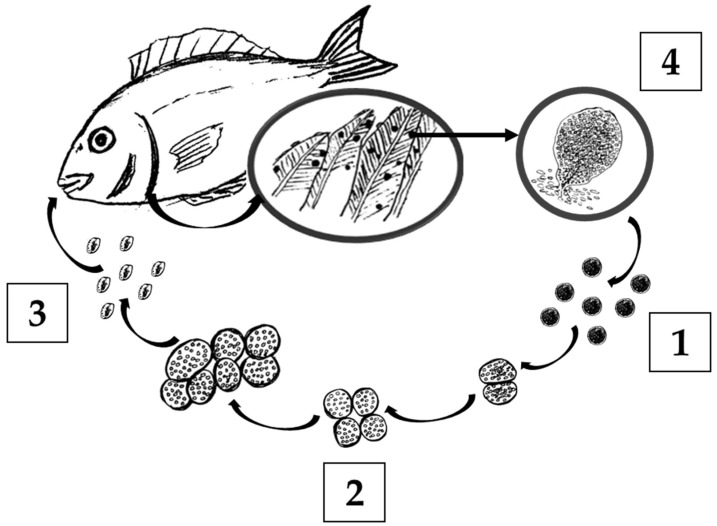
Life cycle of the fish parasite *Amyloodinium ocellatum*, comprising the resting/encysted stage tomont (1), which divides into tomites (2), the dinospore free-living stage (3), and the parasitic stage trophont (4).

**Figure 2 toxins-14-00467-f002:**
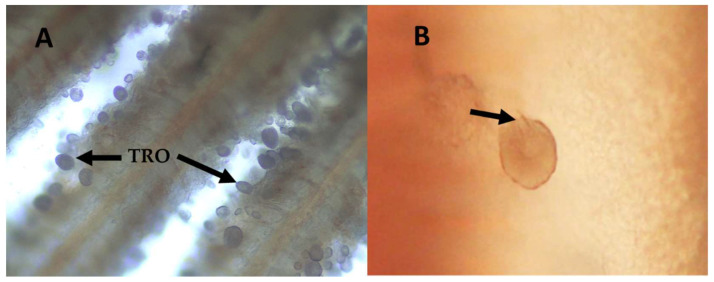
Wet mount of gilthead seabream (*Sparus aurata*) gills at 24 h post-infestation with *Amyloodinium ocellatum.* (**A**) Trophonts (TRO, attached to the gill) are present in the gill at 24–36 h post-infestation (100×); (**B**) Trophont with a visible stomatopod structure inserted into the gill tissue (black arrow) (400×).

**Figure 3 toxins-14-00467-f003:**
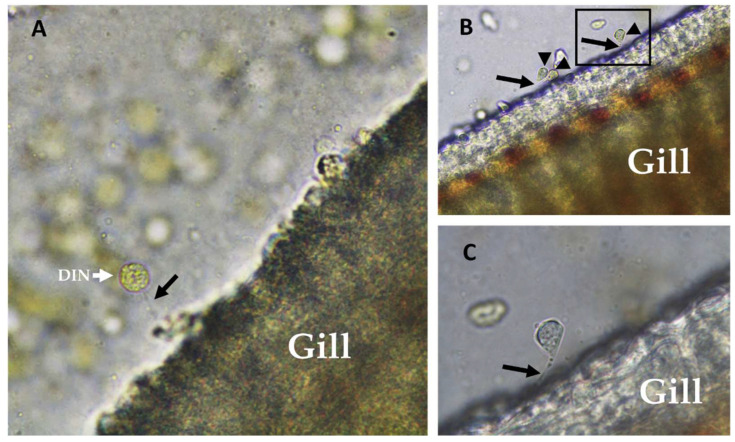
Wet mount of gilthead seabream (*Sparus aurata*) gills at 5 h post-infestation with *Amyloodinium ocellatum.* (**A**) Dinospore (DIN) with a filamentous structure (black arrow) attached to the gill (400×); (**B**) Dinospores presenting with an elongation, characteristic of the transition phase toward the trophont stage (black arrowhead), and the connecting filament (black arrow) (200×); (**C**) Detail of the image in panel B, showing the filamentous structure (black arrow) that connects the dinospore to the gill (400×).

**Figure 4 toxins-14-00467-f004:**
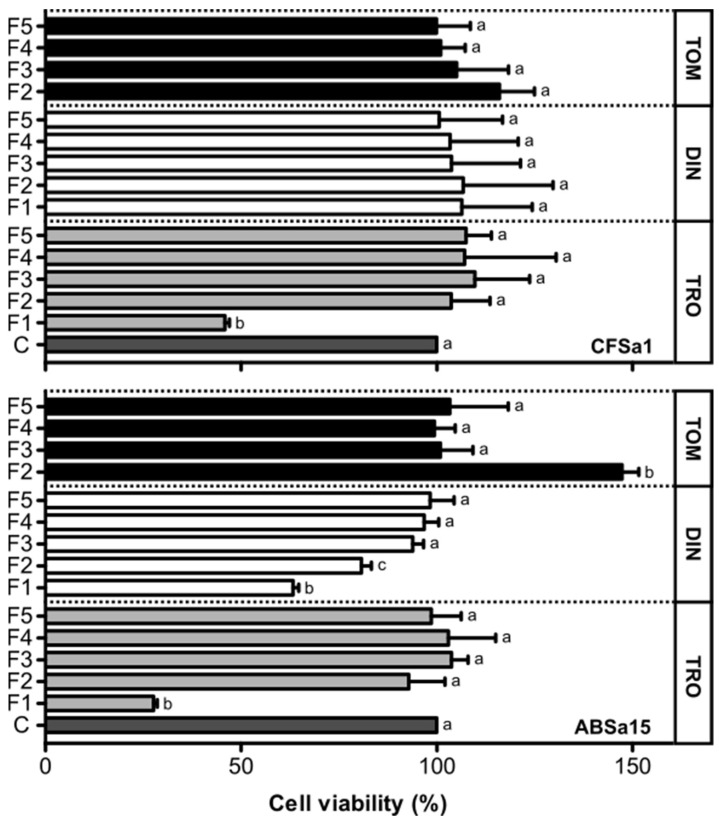
Viability of gilthead seabream (*Sparus aurata*) branchial arch (ABSa15) and caudal fin (CFSa1) cells exposed to fractions of *Amyloodinium ocellatum* life stages crude extract (TOM—Tomonts, DIN—Dinospores, TRO—Trophonts), Control (C); F1 to F5: fractions prepared from the crude extracts using solvents of increasing polarity. Significant differences (*p* < 0.001) were determined through one-way ANOVA followed by the Tukey’s post-test, and the differences between fractions in each life phase are indicated using a–c letters. Values are mean ± standard deviation. N = 3.

**Figure 5 toxins-14-00467-f005:**
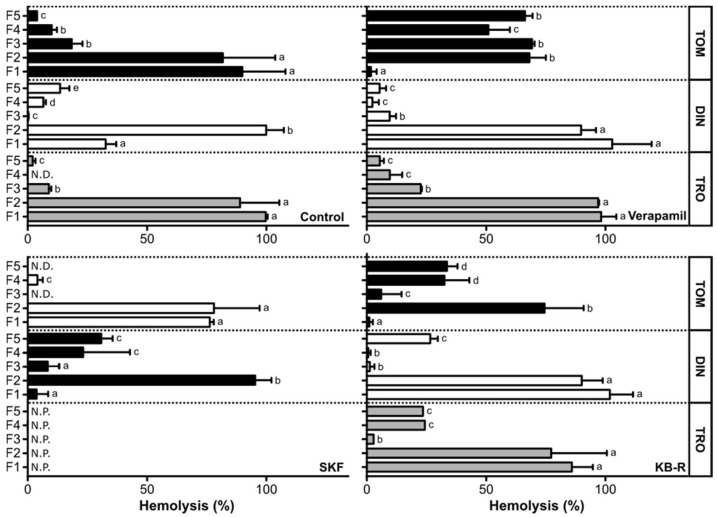
Hemolytic effect of *Amyloodinium ocellatum* tomont (TOM), dinospore (DIN) and trophont (TRO) crude extract fractions prepared using a solvent of increasing polarity. Fractions (F1 to F5) were tested alone (Control) or pre-incubated with Ca^2+^ channel antagonists (verapamil (Verapamil), SKF96365 (SKF) or KB-R7943 (KB-R)) (N.D.—Not detected, N.P.—Not performed). Significant differences (*p* < 0.05) were determined through two-way ANOVA followed by a Bonferroni post-test and the differences between fractions in each life phase are indicated using a–c letters. Values are mean ± standard deviation. N = 2.

**Figure 6 toxins-14-00467-f006:**
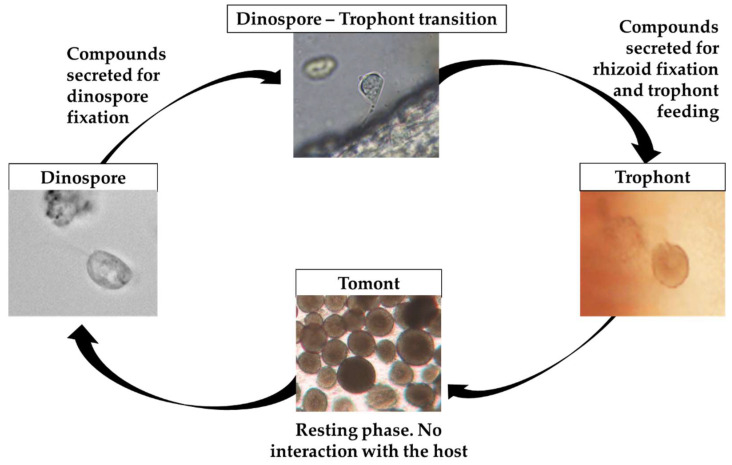
Scheme showing the putative action of the compounds present in the active fractions of *Amyloodinium ocellatum* crude extracts, based on their action on cell viability and hemolytic activity.

## Data Availability

Not applicable.

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
