# Peer review of "Cytotoxic and Hemolytic Activities of Extracts of the Fish Parasite Dinoflagellate Amyloodinium ocellatum"

_toxins, 2022, doi:10.3390/toxins14070467_

Round 1
Reviewer 1 Report
Commercial fish diseases constitute important limitations to aquaculture with significant impact on the economy. Cultured fish are particularly vulnerable in pathogens, whereas toxins are often found in large concentrations in aquaculture units. In this context, the study is of high importance presenting new findings regarding the dinoflagellate A. ocellatum. The study is well organised. I would only suggest the addition of a part in the discussion evaluating the potential risk of the found compounds for Sparus aurata tissues intended for human consumption for the food industry, as well as some minor changes as follows:
Lines 6-8: I do not agree with term “however” because anoxia, osmoregulatory and impairment may in combination with the toxins cause the mortalities, under the concept of confection with bacteria. Maybe rephrase
line 42: they
Lines 45-46: The way it is written it seems to mean that the asymptomatic outbreaks are responsible for the high morbidity and mortality. Please rephrase
Lin line 86 please define specially in which fish species the in vitro cell systems were tested
In Figures 4 and 5 legends please indicate what the “a”, “b” and “c” indicate on the plot. Is it statistical significance at different levels. Please explain in detail
Author Response
Response to Reviewer 1 Comments
Dear reviewer,
We appreciate your comments and suggestions. Above you will find our answers after each comment. We fully addressed all of them and we believe to have now a much better manuscript. We also improved the English on the document, that was revised by an English native. All the changes are tracked in green.
Point 1: Commercial fish diseases constitute important limitations to aquaculture with significant impact on the economy. Cultured fish are particularly vulnerable in pathogens, whereas toxins are often found in large concentrations in aquaculture units. In this context, the study is of high importance presenting new findings regarding the dinoflagellate A. ocellatum. The study is well organised. I would only suggest the addition of a part in the discussion evaluating the potential risk of the found compounds for Sparus aurata tissues intended for human consumption for the food industry.
Response 1: First of all, Thank for your kind comment.
Regarding the addition of a new part on the discussion about the evaluation of the potential risk for the consumer of the found compounds for Sparus aurata tissues intended for human consumption for the food industry, we highly appreciated this reviewer comment. It is our aim to disclose any toxic compound(s) associated/produced by A. ocellatum. Our study suggests, at least in the trophont stage, the presence of compounds with mechanisms of action similar to certain marine toxins. Studies based on LC-MSMS analyses are being performed to investigate and search for marine toxins in A. ocellatum extracts. However, at this moment it is rather speculative to discuss about any possible toxins produced by A. ocellatum which can potentially accumulate in fish and that could then be of risk to human consumption.
Point 2: Lines 6-8: I do not agree with term “however” because anoxia, osmoregulatory and impairment may in combination with the toxins cause the mortalities, under the concept of confection with bacteria. Maybe rephrase .
Response 2: The reviewer’s observation is correct, and text was amended accordingly.
Point 3: Line 42: they
Response 3: The reviewer’s observation is correct, and text was amended accordingly.
Point 4: Lines 45-46: The way it is written it seems to mean that the asymptomatic outbreaks are responsible for the high morbidity and mortality. Please rephrase.
Response 4: The reviewer’s observation is correct, and text was amended accordingly.
Point 5: In line 86 please define specially in which fish species the in vitro cell systems were tested.
Response 5: The reviewer’s observation is correct. The text was amended accordingly to include the fish species from where the cell lines were originated.
Point 6: In Figures 4 and 5 legends please indicate what the “a”, “b” and “c” indicate on the plot. Is it statistical significance at different levels. Please explain in detail.
Response 5: The reviewer’s observation is correct. The legends in Figures 4 and 5 were amended accordingly.

Reviewer 2 Report
General comments:
It is an interesting article, with some grammatical errors that induce the need for a correction made by a native English speaker.
An annoying element for me is the suspicious similarity of some statements in this manuscript with those found in:
Márcio Moreira, Lucía Soliño, José I. Navas, Pedro Rodrigues, Florbela Soares, Pedro Reis Costa, 2019. Is Amyloodinium ocellatum a toxin-producing parasitic dinoflagellate? - a toxicological study. Conference: L’AQUA 2019 - Latin American & Caribbean Aquaculture.
As the authors' information was removed for double-blind peer review, I have no way of knowing if it is plagiarism or not.
Examples of grammatical errors
Line 37: it would be better to start a sentence using the full name of the parasite (Amyloodinium ocellatum) instead of the shortened variant (A. ocellatum);
Line 39: "consists of a", instead of "consists in a";
Line 42: I suppose "they attach" instead of "the attach";
...and so on!
Author Response
Response to Reviewer 2 Comments
Dear reviewer,
We appreciate your comments and suggestions. Above you will find our answers after each comment. We fully addressed all of them and we believe to have now a much better manuscript. We also improved the English on the document, that was revised by an English native. All the changes are tracked in green.
Point 1: It is an interesting article, with some grammatical errors that induce the need for a correction made by a native English speaker.
Response 1: First of all, thank for your kind comment.
Following your comment on the grammatical errors, we asked to a native English speaker to correct our manuscript. We introduced an aknowledgment to this person in the end of the manuscript.
